# Cutting the First Turf to Heal Post-SSRI Sexual Dysfunction: A Male Retrospective Cohort Study

**DOI:** 10.3390/medicines9090045

**Published:** 2022-09-01

**Authors:** Rosaria De Luca, Mirjam Bonanno, Alfredo Manuli, Rocco Salvatore Calabrò

**Affiliations:** 1Neurorehabilitation Unit, IRCCS Centro Neurolesi “Bonino Pulejo”, 98166 Messina, Italy; 2UOC Physical Medicine and Rehabilitation, AOU Policlinico G Martino, 98166 Messina, Italy

**Keywords:** PSSD, pharmacological approach, SSRI

## Abstract

Post-SSRI sexual dysfunction (PSSD) is a set of heterogeneous sexual problems, which may arise during the administration of selective serotonin reuptake inhibitors (SSRIs) and persist after their discontinuation. PSSD is a rare clinical entity, and it is commonly associated with non-sexual concerns, including emotional and cognitive problems and poor quality of life. To date, however, no effective treatment is available. The aim of this study was to retrospectively evaluate the potential efficacy of the different treatments used in clinical practice in improving male PSSD. Of the 30 patients referred to our neurobehavioral outpatient clinic from January 2020 to December 2021, 13 Caucasian male patients (mean age 29.53 ± 4.57 years), previously treated with SSRIs, were included in the study. Patients with major depressive disorder and/or psychotic symptoms were excluded a priori to avoid overlapping symptomatology, and potentially reduce the misdiagnosis rate. To treat PSSD, we decided to use drugs positively affecting the brain dopamine/serotonin ratio, such as bupropion and vortioxetine, as well as other compounds. This latter drug is known not to cause or reverse iatrogenic SD. Most patients, after treatment with vortioxetine and/or nutraceuticals, reported a significant improvement in all International Index of Erectile Function-(IIEF-5) domains (*p* < 0.05) from baseline (T0) to 12-month follow-up (T1). Moreover, the only patient treated with pelvic muscle vibration reached very positive results. Although our data come from a retrospective open-label study with a small sample size, drugs positively modulating the central nervous system serotonin/dopamine ratio, such as vortioxetine, could be used to potentially improve PSSD. Large-sample prospective cohort studies and randomized clinical trials are needed to investigate the real prevalence of this clinical entity and confirm such a promising approach to a potentially debilitating illness.

## 1. Introduction

Selective serotonin reuptake inhibitors (SSRIs) are one of the most used psychiatric drugs, either due to an on-label or to an off-label application [1]. Post-SSRI sexual dysfunction (PSSD) is a set of heterogeneous sexual disorders that may arise during the administration of SSRIs and persist after their discontinuation. PSSD is an iatrogenic, idiosyncratic disorder, as well as a clear example of the post-drug syndromes [2]. It mainly develops following cessation of SSRIs, but other classes of antidepressant drugs have also been reported to cause the disease [3,4]. Tricyclics, serotonin and norepinephrine reuptake inhibitors (SNRIs) as well as antipsychotics have been reported to cause enduring SD, beside the well-known (although rare) long-term effects on the extrapyramidal system, including tardive dyskinesia [5]. Moreover, other non-psychoactive drugs, such as isotretinoin and finasteride, may cause long-lasting genital anesthesia, loss of libido and other SDs. This rare and/or under-reported clinical entity is still not recognized by many specialists in the field. In 2019, only PSSD gained an official recognition after the European Medical Agency concluded that PSSD is a medical condition that persists after discontinuation of SSRIs and SNRIs [2]. This clinical entity is characterized by a wide array of symptoms that may persist for variable periods or even indefinitely [5,6]. In particular, PSSD includes genital anesthesia, anorgasmia, delayed orgasms, ejaculatory dysfunctions and decreased libido that may arise when SSRIs are established and specifically continue when they have been ceased [7,8]. Notably, many patients define it as a “disconnection” between the brain and the genitals [9]. Additionally, growing reports suggest additional non-sexual symptoms including anhedonia, apathy and blunted affect [10]. It is important to differentiate between depression-related SD symptoms and those of PSSD, since some symptoms, such as genital anesthesia, seem to be more associated with PSSD rather than with depression [11,12,13,14]. In fact, depression is strongly associated with SD as part of the core depressive syndrome, in which sexual function may be diminished or absent. In both sexes, decreased sexual desire in depression is the most prominent symptom, and dysfunctions of sexual arousal and orgasm may also occur [15]. There may also be a bi-directional relationship between depression and SD, given that patients who are depressed might not search for sexual intimacy, and, conversely, patients with SD might experience reactive depression [16]. SD and depression are both syndromes that may be caused by the same dysfunctional brain systems. Indeed, the treatment of depression itself may result in iatrogenic SD, as dopamine is known to enhance libido and sexual arousal, whereas serotonin has an inhibitory effect on sexuality [16]. Although SD is a common side effect of SSRIs, its pathophysiology remains largely unknown, as well as therapeutic treatment [17]. No rational or consistent treatment has been found for this disorder [18]. It is imperative for clinicians to be aware of non-sexual symptoms and to be able to differentiate between PSSD-associated SD and depression-related SD, as each of their symptoms can be quite distinctive with a few symptoms overlapping. Consequently, there are no well-designed clinical trials and, therefore, a clear consensus on treatment modalities has not been reached so far.

For these reasons, the aim of this study was to retrospectively evaluate the potential efficacy of the different treatments used in clinical practice in improving male PSSD, and to shed some light on the management of this growing problem.

## 2. Materials and Methods

### 2.1. Study Population and Design

This retrospective cohort study included patients with a diagnosis of PSSD who attended the neurobehavioral outpatient clinic of the IRCCS Centro Neurolesi “Bonino-Pulejo” (Messina, Italy) between January 2020 and December of 2021.

The patients enrolled were referred by general practitioners or other neurologists/psychiatrists to RSC (a recognized specialist of such syndromes), or they reached the clinic through information coming from Google or other media/social sites. They were screened for current iatrogenic or psychogenic SD before PSSD diagnosis was carried out. In fact, the enrolled patients were almost diagnosed and treated for different psychiatric disorders in other outpatient clinics, and came to our attention due to the onset of the symptom after SSRI intake and/or its persistence after withdrawal. Then, a physical examination with a complete sexual hormonal profile (including testosterone, LH, FSH, prolactin and estrogens) as well as a psychosexological assessment (including sexual attitude and habits, previous SD, response to sexual stimuli, libido and fantasies) were performed. Demographic characteristics (sex, age, relationship status, country of origin, occupation and daily activities) and a detailed medical history of all participants were also collected.

Diagnosis of PSSD was based on the existing criteria [19]. In particular, the following “core” inclusion criteria were used:

NecessaryPrior treatment with a serotonin reuptake inhibitor.An enduring change in somatic (tactile) or erogenous (sexual) genital sensation after treatment stops.

Additional3.Enduring reduction in or loss of sexual desire.4.Enduring erectile dysfunction (males).5.Enduring inability to orgasm or decreased sensation of pleasure during orgasm.6.The problem is present for ≥1 month after stopping treatment.

There should be:7.No evidence of pre-drug sexual dysfunction that matches the current profile.8.No current medical conditions that could account for the symptoms.9.No current medication or substance misuse that could account for the symptoms.

Exclusion criteria were: (i) use, abuse or misuse of any drug potentially affecting sexuality; (ii) a clinical history of urologic, endocrine or systemic disease; (iii) severe depression or other concomitant psychiatric disorders, with a Hamilton Rating Scale for Depression < 17, which was administered by a trained psychiatric rehabilitation therapist. In particular, to avoid overlapping symptomatology and reduce the misdiagnosis rate, patients previously diagnosed with major depressive disorder (MDM), bipolar disorder and psychotic symptoms were a priori excluded. Afterwards, the final sample was composed of individuals with anxiety disorders and/or adjustment disorders.

Only patients with a high probability of PSSD according to the previously published criteria [19], as well as the clinical assessment, were included.

Given that no recognized treatment for the syndrome exists, patients were treated based on their features, needs, expectancies and according to the few data coming from case reports and series. Usually, an antidepressant with a dopaminergic/noradrenergic profile or antagonizing/positively modulating the serotonergic system (i.e., with fewer or no known SD side effects) was used, as well as nutraceuticals and/or PDE5 inhibitors.

The Hospital Research Ethical Committee of the IRCCS Neurolesi approved this study (IRCCSME-CE-31/2021), and all participants gave their consensus for data publication.

### 2.2. Outcome Measure

Patients were asked to fill out the International Index of Erectile Function-15 (IIEF) [20], a well-validated psychometric tool to measure sexual function (with regard to erection) at baseline and after a follow-up period of 12 months. The IIEF-15 is a multidimensional, self-administered investigation that has been found to be useful in the clinical assessment of erectile dysfunction (ED) and treatment outcomes in clinical trials. It has been recommended as a primary endpoint for clinical trials of ED and for diagnostic evaluation of ED severity [21]. A score of 0–5 is awarded to each of the 15 items that examine the 4 main domains of male sexual function: erectile function, orgasmic function, sexual desire and intercourse satisfaction. The questionnaire was administered in Italian. The IIEF-15 scoring ranges from 0 to 30: a score of 1–10 is indicative of severe ED; 11–16 moderate ED; 17–25 mild ED; and 26–30 absence of ED.

### 2.3. Statistical Analysis

Mean, standard deviation, median lowest, highest, frequency and ratio values were used in the descriptive statistics of the data. The distribution of the variables was measured by the Kolmogorov–Smirnov test. Due to the non-normal distribution of the study variables and the small population, we performed the Wilcoxon signed-rank test in the analysis of the dependent quantitative data, using software R 4.1.3 (Messina, Italy) [22]. A *p*-value < 0.005 was established.

## 3. Results

Of the 30 patients referred to our neurobehavioral outpatient clinic from January 2020 to December 2021, 13 Caucasian male patients, with a mean age of 29.53 (±4.57) years, were included in the study. The main sociodemographic and clinical characteristics are shown in Table 1.

Less than half of the patients only complained of SD (with anorgasmia and loss of libido being the most frequent ones), whilst in about 23%, SD was associated with cognitive problems, in 8% emotional problems and in about 25% both cognitive and emotional concerns. The enduring SD was caused only by an SSRI, and those with a more selective profile (i.e., citalopram and escitalopram) were the most common (Table 2).

Notably, different strategic treatments were used to overcome PSSD, with vortioxetine being the most common and effective one. After the various treatments, the IIEF-15 score improved significantly (*p* > 0.05) in the majority of the sample, except for two cases, one treated with vortioxetine and nutraceuticals and the other with bupropion, tadalafil and a nutraceutical. At T0, nearly all PSSD patients treated with vortioxetine (10–20 mg according to each patient’s response) started from a severe level of SD, according to the IIEF-15. At T1, we observed a significant improvement in the IIEF score, with a substantial reduction in SD (with regard to anorgasmia), achieving a high percentage of therapeutic success (from 33.3 to 60%) (see Table 3). Therefore, most of the patients (10/12) reported an improvement in the main sexual and non-sexual symptoms as per the IIEF score, except two cases in which the therapeutic success rate was equal to 0% where we did not find any score increase. Moreover, in the only drug-resistant patient receiving Vibra-Plus Therapy, we observed an improvement of 50% in the IIEF-15 score at T1.

## 4. Discussion

This real-life retrospective study describes a small cohort of patients diagnosed with PSSD according to the recently published selection criteria [19] and treated with strategic pharmacological and non-pharmacological interventions. Sexual dysfunction can appear while on treatment and persist after discontinuing any serotonin-reuptake-inhibiting drug [23]. There is a growing awareness that a substantial number of medicines have either positive or negative effects on sexual functioning [24]. These include antibiotics, antihypertensives, lipid-lowering agents, medicines affecting endocrine systems and others [24]. Notably, psychotropic drugs, targeting serotonin and dopamine pathways, are widely recognized as the main drugs responsible for SD. The treatment approaches adopted to overcome iatrogenic SD have been largely aimed at reversing the acute sexual effects rather than reversing the mechanism that leads to enduring effects [25]. Our preliminary data advance the research in the management of PSSD, as the clinical use of vortioxetine, as well as bupropion (although in fewer cases), which is associated with nutraceuticals, might be considered as a potentially effective treatment of this enduring problem. In fact, in nearly all patients treated with these strategic interventions, we observed a positive change in the level of SD, according to the IIEF-15 score as well as an improvement in non-sexual side effects (evaluated by a specific psychosexological in an interview). However, in the current literature, there is still no definitive treatment for PSSD. Some authors suggest that a treatment option for patients might be to take bupropion or nefazodone, which are antidepressants that are known to cause few or no sexual adverse effects [25]. In fact, bupropion does not have serotonergic activity and, hence, does not affect sexual function in patients. According to the literature, patients treated with bupropion report less SD, and also document a recovery in satisfaction, desire and frequency of sexual activity [26], given that the drug has a positive effect on dopaminergic pathways. In line with our results, Jacobsen et al. (2015) showed that switching antidepressant therapy to vortioxetine may be beneficial for patients experiencing SD during antidepressant therapy with SSRIs [27].

Vortioxetine has been approved for the treatment of adults with major depressive disorders (MDDs) since 2013, and subsequently it has been shown that the drug may be particularly beneficial for specific populations of patients, including those with treatment-emergent SD and patients experiencing certain cognitive symptoms [28,29]. This is possible because of the multimodal action of vortioxetine; indeed, it is a serotonin (5-HT) transporter inhibitor that also acts on several 5-HT receptors, such as the 5-HT3 and 5-HT1A receptors [30]. This is why the drug might have led to such positive results in our sample, even if the patients were affected by anxiety/adjustment disorders. One may be concerned that treatment with bupropion or vortioxetine is more likely to treat ongoing depression, including symptoms of SD, anhedonia, apathy, cognitive symptoms and emotional blunting, than to reverse a postulated effect of an SSRI after discontinuation [31]. Nonetheless, we believe that SD and related problems were more likely due to the enduring iatrogenic effect, since our patients were properly assessed at baseline, and diagnosed with PSSD according to the current available criteria [19] and after an accurate psychosexological anamnesis.

The use of turmeric could have positive effect on sexual function in some cases, since it is known to help increase BDNF and reduce inflammation, and thus also improve depression [32]. Moreover, the compound we used might have also acted on mood and anxiety according to the well-known bi-directional relationship between sexuality and depression [33]. Men’s sexual functioning could be improved by the use of nutraceuticals, as they may increase libido and genital arousal, and may be considered as an alternative treatment of PSSD. In a previous case report by Calabro’ et al., a dietary supplement called EDOVIS has been used to restore PSSD [34]. It is composed of L-Citrulline, tribulus terrestris, andean maca, damiana, muira puama, and folic acid, which are useful for the physiological sexual activity of males [35]. Today, nutraceutical and functional food components could also represent a strategic approach to treat SD, according to a holistic approach [36]. The main component of the nutraceutical, i.e., nitric oxide—NO, is the pivotal factor involved in the endothelium-dependent relaxation of the human corpus cavernosum, potentially boosting erectile function and genital sensation [37]. Nutraceuticals and dietary supplements are an accessible alternative that men with ED use to attempt to address their SD, as reported in a recent review [38]. In particular, the main nutraceuticals included a series of natural components: ginseng, composed of biologically active compounds called ginsenosides and ginseng saponins [39]; the amino acid L-arginine, which is a precursor to NO and is converted by NO synthase [40]; Tongkat Ali, an aphrodisiac herbal extract, because of its ability to increase testosterone levels [41]; horny goat weed, whose bioactive ingredient is icariin, which has historically been used as an aphrodisiac and herbal treatment for ED in Chinese men [42]; tribulus terrestris, an herbal plant that has been claimed to improve physical performance and sexual activity [43]; Maca, a vegetable derived from the Lepidium meyenii plant that has been historically used as both a nutritional supplement and fertility enhancer [44]; zinc, a mineral able to improve erectile function [45]; and damiana (also known as turnera diffusa), a well-regarded aphrodisiac ingredient that stimulates sexual desire and performance [46].

Based on such data, our patients were treated with EDOVIS, with some positive results when used alone or in combination/after other compounds. As is known, the market for dietary supplements and nutraceuticals taken to improve the sexual health or psychological well-being of the customer is enormous. However, after accidental and excessive intake of these supplements, some side effects, such as nausea, diarrhea, vomiting and cramping abdominal pain, have been reported [47]. More attention to adverse effects and potential interactions is needed in order to prevent pharmacological interactions and potentially serious medical outcomes.

Waldinger et al. reported the effect of physical therapies, such as low-power laser irradiation, or phototherapy, directed toward the scrotal skin and the shaft of the penis in a male patient with PSSD and penile anesthesia, alleviating anejaculation and erectile dysfunction symptoms of PSSD in the same patient [48]. In our sample, a patient with no response to previous pharmacological treatment received an intensive alternative treatment using Vibra -Plus, with a beneficial role in his sexual symptoms, including sexual hypoesthesia and anorgasmia. In particular, muscle vibrations (MVs) have already been used to manage different pelvic floor dysfunctions due to diverse pathologies [49]. There is converging evidence that MV provides the central nervous system with strong proprioceptive inputs that reach the sensorimotor cortices. This may help to modify the corticospinal excitability, to favor intracortical inhibitory systems and to induce better muscle synergy patterns by acting on the excitability of spinal motoneurons and interneurons. MV may directly act at the spinal level, reducing abnormalities of the spinal excitability and restoring abnormal reciprocal and presynaptic inhibition mechanisms [50], also leading to an improvement in genital sensation. Instead, in the peripheral nervous system, the improvement in erection may depend on the effects of MV on the specific properties of the muscles and surrounding connectivity tissues (including viscoelasticity), as well as on vessel vasodilatation [51]. Moreover, the use of MV plus other kinds of non-invasive neuromodulation could be taken into consideration as a future and promising treatment, as demonstrated by previous works [52]. Different evidence managing transcranial magnetic stimulation demonstrated that focal MV increases or decreases motor-evoked potential amplitude and short intracortical inhibition strength in the vibrated muscles, while opposite changes occur in the neighboring muscles [53]. In this way, pelvic MV may contribute to regulating the contraction and excitability dynamics of the pelvic floor muscles involved in erection.

The presence of few not validated approaches highlights the difficulty in choosing the treatment that must be targeted at the individual level in patients with PSSD. Furthermore, the psychological and behavioral component cannot be underestimated in this kind of patient. Indeed, cognitive-behavioral therapy also has been used by psychiatrists to help patients reach a better understanding of their condition and cope better with their situation. Cognitive-behavioral therapy is useful for dealing with the negative thoughts that develop in many patients, such as sexual inadequacy and low self-esteem [54,55]. Partners need to be involved in this approach because they are collaterally affected by PSSD. Sex therapy and couples counseling should aim to inform the partners that the sexual dysfunction is a side effect of the medication and not a lack of interest. In addition, such behavioral therapies can provide emotional and psychological support for patients and partners [55].

We are aware that the diagnosis of PSSD is not currently recognized by the DSM-V, and that its prevalence is not known because of the lack of well-designed studies. Indeed, depression is frequently associated with SD in both men and women. Clinicians should consider the bi-directional association between depression and SD. Patients reporting SD should be screened for depression, whereas patients presenting with symptoms of depression should be routinely assessed for SD [56]. However, PSSD appears to be a different clinical entity, as recognized by the European Medical Agency in 2019 and current criteria in a consensus paper [19]. Moreover, none of the patients had MDD (an exclusion criterion), which may have accounted for the enduring sexual and affective symptomatology more than adjustment and anxiety disorders. Unfortunately, given that diagnosis was performed before our assessment for PSSD and the stressors were identified and managed by other clinicians, we are not completely able to rule out their potential role in iatrogenic SD.

In addition, it is not possible to ignore that young people without any history of depression or use of antidepressants, trauma or anxiety frequently present with SD, and no cause can be identified. No prospective systematic study starting patients with a psychiatric syndrome, such as MDD, on SSRIs and following persistence of SD or relapse of the symptomatology after medication discontinuation has been carried out, nor will it be possible given the rarity of this complaint. This important issue should be addressed by future studies to better understand this “new” clinical entity and the subtending pathophysiological mechanisms.

Moreover, we have described and treated patients with suspected iatrogenic PSSD independently of their psychiatric diagnosis, although this can make the sample less homogeneous. Furthermore, it could be useful to report in futures works the correlation between SSRI and testosterone levels, since abnormalities in sexual hormonal levels have been reported after the drug intake [57]. Nonetheless, our patients’ sexual hormones at assessment were within the normal range.

The study had some other limitations. First of all, the retrospective design prevented us from developing any a priori hypothesis. However, this was an open-label observational study performed in a real-life context that could be the basis for planning future randomized clinical trials. The small sample size was another limitation, but it is not so easy to collect larger homogeneous cohorts as the disease is still unrecognized and few papers exist to guide the right diagnosis. Other outcome measures, such as the Patient Health Questionnaire-9 and the Generalized Anxiety Disorders 7-item scale at baseline and after intervention as well as assessment for change over time, would have been more helpful than just the initial HAMD score to achieve the diagnosis and assess whether and to what extent improvement could have continued. However, we administered this scale to exclude patients with severe depression/somatization symptoms at assessment, beyond those who were a priori excluded due to being affected by MDD. Future trials should address this important issue, and more systematic large-sample cohort studies on patients on SSRIs are necessary to investigate the “real” prevalence of PSSD.

Moreover, since this was a retrospective small-sample real-life study, it was not possible to compare the efficacy of the different compounds, either used alone or in combination, in improving PSSD. This is why we have only reported a single patient’s therapeutic success rate, but larger studies are needed to solve this important issue so as to give indications on the best therapeutic approach. Finally, a pharmacogenetic assessment to see if the patients were slow metabolizers might have been helpful to better understand the cause of the clinical entity.

As a strength, our sample was homogeneous, as in all of the patients’ PSSD was caused by an SSRI, and diagnosis was made after a strict application of the ongoing criteria and an accurate psychosexological anamnesis and clinical investigation.

## 5. Conclusions

As far as we know, this is the first study that attempted to identify therapeutic intervention strategies for enduring sexual dysfunction related to the use of SSRIs. Although our data come from a retrospective open-label study with a small sample size, drugs positively modulating the central nervous system’s serotonin/dopamine ratio, such as vortioxetine, could be used to potentially improve PSSD. Larger randomized clinical trials are needed to confirm our data and find promising neuropharmacological approaches to better manage this potentially debilitating illness.

## Figures and Tables

**Table 1 medicines-09-00045-t001:** Sociodemographic and clinical variables of the PSDD sample.

Patients	13
Age (years)	29.53 (±4.57)
Education	
Elementary school	0
Middle school	0
High school	11 (84.61)
University	2 (15.39)
Etiology	
Citalopram	3 (23.07)
Paroxetine	3 (23.07)
Sertraline	3 (23.07)
Fluoxetine	1 (7.72)
Escitalopram	3 (23.07)
Diagnosis before PSDD	
Adjustment disorder	7 (53.86)
Adjustment disorder associated with anxiety disorder	1 (7.69)
Adjustment disorder with an anxiety disorder with panic attacks	1 (7.69)
Generalized anxiety disorder with panic attacks	2 (15.38)
Obsessive compulsive disorder	2 (15.38)

**Table 2 medicines-09-00045-t002:** SSRIs taken by the patients, treatment duration and time since PSSD onset.

Patient No.	SSRIs	Dose (mg)	Treatment Duration (Months)	Onset of Enduring Sexual Side Effects
1	Citalopram	20	3	4 weeks after treatment discontinuation
2	Paroxetine	20	1	3 weeks after starting treatment
3	Sertraline	100	12	3 weeks after discontinuation
4	Citalopram	20	48	2 weeks after starting treatment
5	Paroxetine	40	4	3 weeks after starting
6	Fluoxetine	20	8	4 weeks after starting treatment
7	Escitalopram	10	3	2 weeks after starting treatment
8	Sertraline	20	5	3 weeks after starting treatment
9	Paroxetine	40	12	4 weeks after starting treatment
10	Citalopram	20	12	2 weeks after withdrawal
11	Sertraline	20	3	4 weeks after discontinuation
12	Escitalopram	20	6	2 weeks after starting treatment
13	Escitalopram	10	8	2 weeks after discontinuation

**Table 3 medicines-09-00045-t003:** Statistical analysis of IIEF-15 scores, with each patient’s raw score from T0-T1, treatment used and percentage of therapeutic success.

Before Treatment (IIEF—% Mean Score T0)			After Treatment (IIEF—Mean Score T1)					*p*-Value
Mean	SD	Median	Mean	SD		Median		0.003
7.3	1.84	7	17.7	6.01		19		
Strategic Treatments		Type and dose (mg)		IIEF-15 score		Level of sexual dysfunction		Percentage of therapeutic Success
				T0	T1	T0	T1	
Pharmacological		Vortioxetine (10)		8	22	Severe	Mild	46.66%
				6	23	Severe	Mild	56.66%
				7	22	Severe	Mild	50%
				7	25	Severe	Mild	60%
				5	15	Severe	Moderate	33.3%
		Vortioxetine (20) and tumeric		7	16	Severe	Moderate	30%
		Vortioxetine (15) and nutraceuticals		11	11	Moderate	Moderate	0%
		Bupropion (300)		9	12	Severe	Moderate	10%
		Bupropion (150), tadalafil (10 and nutraceuticals		5	5	Severe	Severe	0%
		Nutraceuticals and bupropion (150)		6	19	Severe	Mild	43.33%
		Nutraceuticals		6	15	Severe	Moderate	30%
		tadalafil (10)		8	20	Severe	Mild	40%
Non-Pharmacological		Vibra-Plus		10	25	Severe	Mild	50%

## Data Availability

Data are available on request from the corresponding author.

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
