# Peer review of "Cutting the First Turf to Heal Post-SSRI Sexual Dysfunction: A Male Retrospective Cohort Study"

_medicines, 2022, doi:10.3390/medicines9090045_

Round 1
Reviewer 1 Report
Much needed review to address medication-related issues commonly encountered in clinical practice.
Would recommend taking a more evidence-based approach to the analysis of reviewed studies.
There is no compelling evidence to promote Vortiotexine as a solution for PSSD and there seems to be some bias about highlighting it several times in your paper.
Would suggest a more in-depth review of nutraceuticals and discuss each ingredient rather than packaged brand.
Overall, great idea and an important topic, but the manuscript needs to be improved.
Author Response
- Much needed review to address medication-related issues commonly encountered in clinical practice.
Added, as suggested. antipsychotics, other antidepressants than SSRI and finasteride may cause the syndrome.
- Would recommend taking a more evidence-based approach to the analysis of reviewed studies.
Done
- There is no compelling evidence to promote Vortiotexine as a solution for PSSD and there seems to be some bias about highlighting it several times in your paper.
This issue has been addressed and tuned down, only suggesting to try neuropharmacological modulation, including vortioxetine, to face Post SSRI syndrome
- Would suggest a more in-depth review of nutraceuticals and discuss each ingredient rather than packaged brand.
Done.
Overall, great idea and an important topic, but the manuscript needs to be improved.
We have improved the paper as suggested.
Reviewer 2 Report
I recognize this is a retrospective study, but it conflates sexual dysfunction (SD) associated with major depressive disorder (MDD), with side effects of medications, and with SD due to other causes e.g. trauma, medical conditions, etc.
Here is my critique.
1. The postulated post-SSRI sexual dysfunction (PSSD) is not a disease, and is not a recognized diagnosis in DSM-5 or ICD-11. The description provided is actually consistent with SD as a symptom of MDD or SD with SRI treatment, particularly SD (70% of patients with MDD have SD as a symptom of the depression), anhedonia, apathy, blunted affect, etc. Ways to delineate this include using a patient reported outcome (PRO) measure for MDD like the PHQ-9, the GAD-7 for anxiety, which also contributes/causes the symptoms above and can be part of depressive symptomatology or is comorbid in 50% of patients with MDD, and perhaps the Oxford Depression Questionnaire to evaluate for emotional blunting.
2. Young people without any history of depression or use of antidepressants, trauma or anxiety frequently present with SD, and no cause can be identified. No prospective systematic study when starting patients with MDD on SSRIs and following for persistence of SD or relapse of MDD after medication discontinuation has been done, nor will it be able to be done given the rarity of this complaint.
3. It looks like only 6 of the 13 patients had a depressive diagnosis. I am not sure what a reactive depression is – perhaps it is an adjustment disorder related to an environmental stressor rather than MDD or the stressor was a trigger for symptoms of MDD. That stressor needs to be identified and determined if still ongoing (chronic and contributing to SD after med discontinuation). Why were the other 7 patients put on SSRIs? What were their diagnoses? The antidepressants could have been used for an anxiety disorder, but that makes this a heterogeneous group.
4. In Table 2, many patients have the designation about onset of “During the drug intake.” What does that mean?
5. Treatment with bupropion or vortioxetine (antidepressants with low rates of SD) is more likely to have treated ongoing depression including symptoms of SD, anhedonia, apathy, cognitive symptoms and emotional blunting than to reverse a postulated effect of an SSRI after discontinuation.(they are both used as an option for switching from SSRI that is causing SD). I do not know what KuKuma is. Is that Kumkuma (turmeric)? Tumeric/curcumin is also reported to help increase BDNF and reduce inflammation, and thus improve depression. As far as I know, it does not help with sexual dysfunction. I am not aware that adding NO helps with SD/ED although a PDE-5 inhibitor might.
6. It is reported that “a complete sexual hormonal profile” was done, but neither the specific tests performed nor the results are reported. SSRIs are known to reduce testosterone, and that may be relevant to these complaints.
7. Suggest if you cannot get validated measures of each patient’s depression and anxiety symptoms, that the above issues be used to change the focus and orientation of the paper.
Author Response
Here is my critique.
- The postulated post-SSRI sexual dysfunction (PSSD) is not a disease, and is not a recognized diagnosis in DSM-5 or ICD-11. The description provided is actually consistent with SD as a symptom of MDD or SD with SRI treatment, particularly SD (70% of patients with MDD have SD as a symptom of depression), anhedonia, apathy, blunted affect, etc. Ways to delineate this include using a patient reported outcome (PRO) measure for MDD like the PHQ-9, the GAD-7 for anxiety, which also contributes/causes the symptoms above and can be part of depressive symptomatology or is comorbid in 50% of patients with MDD, and perhaps the Oxford Depression Questionnaire to evaluate for emotional blunting.
We understand this concern, and agree that the diagnosis is not recognized by the DSMV. It is known that SD is associated with MDD, nonetheless PSDD seems to be a different entity, as better specified in the text with criteria recognized by some authors/consensus. Indeed, patients in the group were almost diagnosed and treated for different psychiatric disorders in other Outclinics, and came to our observation for the onset of the symptom after SSRI intake and its persistence after the withdrawal.
- Young people without any history of depression or use of antidepressants, trauma or anxiety frequently present with SD, and no cause can be identified. No prospective systematic study when starting patients with MDD on SSRIs and following for persistence of SD or relapse of MDD after medication discontinuation has been done, nor will it be able to be done given the rarity of this complaint.
We have added this important comment in discussion since we agree with this concern
- It looks like only 6 of the 13 patients had a depressive diagnosis. I am not sure what a reactive depression is – perhaps it is an adjustment disorder related to an environmental stressor rather than MDD or the stressor was a trigger for symptoms of MDD. That stressor needs to be identified and determined if still ongoing (chronic and contributing to SD after med discontinuation). Why were the other 7 patients put on SSRIs? What were their diagnoses? The antidepressants could have been used for an anxiety disorder, but that makes this a heterogeneous group.
We have described and treated patients with suspected iatrogenic PostSSRI syndrome independently of their psychiatric diagnosis. However, this important issue has been added, ans also better clarify “reactive depression”
- In Table 2, many patients have the designation about onset of “During the drug intake.” What does that mean?
It means that the Sd appears while on medication and persisted after discontinuation as per inclusion and consensus criteria diagnosis
- Treatment with bupropion or vortioxetine (antidepressants with low rates of SD) is more likely to have treated ongoing depression including symptoms of SD, anhedonia, apathy, cognitive symptoms and emotional blunting than to reverse a postulated effect of an SSRI after discontinuation.(they are both used as an option for switching from SSRI that is causing SD).
I do not know what KuKuma is. Is that Kumkuma (turmeric)? Tumeric/curcumin is also reported to help increase BDNF and reduce inflammation, and thus improve depression. As far as I know, it does not help with sexual dysfunction. I am not aware that adding NO helps with SD/ED although a PDE-5 inhibitor might.
Although the referee is right that SD is often related to depression (see comment on bidirectional relationship), in this case we believe that patients were affected by PSSD. Indeed, severe depression and other psychiatric disorders were an exclusion criteria, as better specified by the HDRS and diagnosis of PSSD done based on available criteria.
More information to explain nutraceuticals effects on SD has been added.
- It is reported that “a complete sexual hormonal profile” was done, but neither the specific tests performed nor the results are reported. SSRIs are known to reduce testosterone, and that may be relevant to these complaints.
We have specified this concern.
- Suggest if you cannot get validated measures of each patient’s depression and anxiety symptoms, that the above issues be used to change the focus and orientation of the paper.
The paper is a retrospective one, suggesting the idea to find a potential treatment for a syndrome which is being recognized by the scientific community. Although we understand the reviewer's concerns, if interpreted with caution, the paper’s result could be a basis for future research. Finally HDRS was used to screen patients with severe depression and related disorders (see exclusion criteria), further helping us maintain the focus of the paper.
Round 2
Reviewer 1 Report
Very practical and important topic relevant to everyday practice.
Excellent ideas and approach. However, the small sample size and lack of consistent results are two major limitations of this study. For example, 0% efficacy in vortioxetine plus nutraceuticals compared to much better efficacy of each modality alone is confusing. Study results description with a more elaborate potential explanation would be desirable. Listing neutraceuticals and potential benefits are welcome by the reading audience. However, some allusion to potential side effects would be a useful disclaimer.
Author Response
Very practical and important topic relevant to everyday practice.
Excellent ideas and approach. However, the small sample size and lack of consistent results are two major limitations of this study. For example, 0% efficacy in vortioxetine plus nutraceuticals compared to much better efficacy of each modality alone is confusing. Study results description with a more elaborate potential explanation would be desirable.
Thank you for this important comment, we have now added more information in results to make them more clear and some better explanations in discussion, where we have specified that
“ Since this is a retrospective small sample real life study, it was not possible to compare the efficacy of the different compounds, either used alone or in combination. This is why we have just reported the single patient’s therapeutic success rate, but larger studies are needed to solve this important issue”.
Listing nutraceuticals and potential benefits are welcome by the reading audience. However, some allusion to potential side effects would be a useful disclaimer.
This information has been added.
Reviewer 2 Report
The abstract was not modified. The introduction reads as if there is no understanding of the rareity of the condition (not a disorder) called PSSD, which will not be able to be proven in a RCT given the billions of people treated with SSRIs and the very small number presenting with these symptoms/given this dx. Also, the complaints are provided as self-identified reports, which are not a valid measure of SD. And they suggest that it is related to tardive dyskinesia. TD is a rare event when related to onset after use of SSRIs. Although the authors indicated in the exclusion criteria "with a Hamilton Rating Scale for Depression<17", I think they meant that a HAMD score of >/= 17 would be exclusionary. It seems curious that urologists would administer a HAMD as it is a clinician rated scale, so I wonder who did interview the patients for the HAMD? A trained psychologist? Something like the PHQ-9 at baseline and after intervention (perhaps more than one subsequent follow-up measurement) and assessment for change over time would have been far more helpful than just an initial HAMD score. Also a measure of anxiety like the GAD-7 would have shed some light on this issue. since anxiety disorders were so prominent in this cohort. It is also curious that none of the patients actually had a diagnosis of major depressive disorder, moderate as the condition for which they received the SSRI (equal numbers of Adjustment disorder or an anxiety disorder) which might mean several things (e.g. diagnosis was downgraded by the diagnosing clinician and/or was a reaction to a situational stressor (especially if that stressor has become chronic), their symptoms prior to entering the study were exacerbated by the anxiety disorder as such patients are far more likely to experience symptoms of their anxiety with medication initiation that they attribute to side effects of medications especially if started at higher doses), etc. The authors still seem to reject the idea that these complaints could all be symptoms of depression or other psychiatric conditions. This is an absolutely critical point and suggests a limited understanding of MDD and its treatment, especially with antidepressants. No corrections were made to the column in Table 2 reflecting "Onset of enduring sexual side effects" for those indicating onset "During the SSRI treatment" as this is by definition required. Was it early or late in the treatment? Did they have SD with their psych dx that persisted through Tx or was the onset noted late in treatment (some of the cases say after 1 month or some other time frame). A pharmacogenetic assessment to see if they were slow metabolizers might have been helpful. It appears only the IIEF was used to measure SD which doesn't include some of the symptoms indicated to be part of PSSD e.g. genital anesthesia, emotional blunting, apathy/amotivation, etc. and which presumably were assessed by patient report. The doses of the prescription drugs, vortioxetine, bupropion, tadalafil are not listed - these are critical, and I suggested the neutraceuticals be described in a footnote, not the text - were these pharmaceutical grade products? The study had potential, but the interpretations of the findings are overly positive and only come from one perspective. It would be helpful to know that patients with these complaints might respond to an antidepressant with an alternative MOA; but that would not support the diagnosis of PSSD which unfortunately seems to be the main interest of the authors.